# SETDB1 enables development beyond cleavage stages by extinguishing the MERVL-driven two-cell totipotency transcriptional program in the mouse embryo

Tie-Bo Zeng[1†‡], Zhen Fu[2†], Mary F Majewski[3], Ji Liao[1], Marie Adams[3], Piroska E Szabó[1]*

[1]Department of Epigenetics, Van Andel Institute, Grand Rapids, United States; [2]Bioinformatics and Biostatistics Core, Van Andel Institute, Grand Rapids, United States; [3]Genomics Core, Van Andel Institute, Grand Rapids, United States

*For correspondence: piroska.szabo@vai.org

[†]These authors contributed equally to this work

Present address: [‡]University of Chicago, Chicago, United States

Competing interest: The authors declare that no competing interests exist.

## eLife Assessment

This study presents a **valuable** finding on maternal SETDB1 as a key chromatin repressor that shuts down the 2C gene program and enables normal mouse embryonic development. The evidence supporting the claims of the authors is **solid**, although the inclusion of a causality test, a mechanistic understanding of SETDB1 targeting, and phenotypic quantification would have greatly strengthened the study. The work will be of broad interest to biologists working on embryonic development, stem cells and gene regulation.

**Abstract** Loss of maternal SETDB1, a histone H3K9 methyltransferase, leads to developmental arrest prior to implantation, with very few mouse embryos advancing beyond the eight-cell stage, which is currently unexplained. We genetically investigate SETDB1's role in the epigenetic control of the transition from totipotency to pluripotency—a process demanding precise timing and forward directionality. Through single-embryo total RNA sequencing of two-cell and eight-cell embryos, we find that *Setdb1*[mat-/+] embryos fail to extinguish one-cell and two-cell transient genes—alongside persistent expression of MERVL retroelements and MERVL-driven chimeric transcripts that define the totipotent state in mouse two-cell embryos. Comparative bioinformatics reveals that SETDB1 acts at MT2 LTRs and MERVL-driven chimeric transcripts, which normally acquire H3K9me3 during early development. The dysregulated targets substantially overlap with DUXBL-responsive genes, indicating a shared regulatory pathway for silencing the two-cell transcriptional program. We establish maternal SETDB1 as a critical chromatin regulator required to extinguish retroelement-driven totipotency networks and ensure successful preimplantation development.

## Introduction

SETDB1 is a histone 3 lysine 9 (H3K9) di- and tri-methyl transferase (*Yang et al., 2002*). In the mouse embryo, it localizes to the inner cell mass of the blastocyst (*Cho et al., 2012*) and is essential for post-implantation development (*Dodge et al., 2004*). Homozygous knockout of *Setdb1* results in embryonic lethality between 3.5 and 5.5 days post coitum (dpc) and prevents the derivation of embryonic

stem cells (ESCs). SETDB1 is required for the silencing of developmental gene promoters (*Bilodeau et al., 2009*). Although *Setdb1*[-/-] zygotic mutants retain maternally deposited SETDB1 protein from the oocyte, conditional knockout of *Setdb1* during oogenesis produces an earlier and more severe maternal-effect phenotype: most oocytes arrest during meiosis, and the few that are fertilized typically fail to develop beyond the two-cell (2c) or eight-cell (8c) stages and never reach the blastocyst (*Eymery et al., 2016*; *Kim et al., 2016*). The basis of this cleavage-stage arrest in the *Setdb1*[mat-/+] embryos remains unknown.

This developmental block coincides with the transition from totipotency to pluripotency, prompting us to investigate whether SETDB1 regulates this critical process. Following fertilization, parental epigenomes undergo extensive reprogramming to enable zygotic genome activation (ZGA). In mice, minor ZGA occurs at the late one-cell (1c) stage and major ZGA at the 2c stage—both essential for development (*Abe et al., 2018*). The 2c stage marks the onset of totipotency, while from the four-cell (4c) stage onward, blastomeres begin committing to either the pluripotent inner cell mass or the trophectoderm lineage (*Piotrowska-Nitsche et al., 2005*). In ESCs, dedifferentiation to two-cell-like cells (2CLCs) (*Macfarlan et al., 2012*) depends on DUX, a pioneer transcription factor that activates ZGA genes (*De Iaco et al., 2017*). Additional ZGA drivers include OBOX, NFYA, NR5A2, and SRF (*De Iaco et al., 2017*; *Iaco et al., 2020*; *Gassler et al., 2022*; *Hermant et al., 2025*; *Ji et al., 2023*; *Lu et al., 2016*), which open chromatin and activate transcription in a combinatorial or compensatory fashion (*Kojima et al., 2025*). Although DUX and individual OBOX proteins are dispensable for ZGA in embryos (*Chen and Zhang, 2019*; *Iaco et al., 2020*; *Guo et al., 2019*; *Ji et al., 2023*), their combined loss (*Ji et al., 2023*), or loss of both DUX and OBOX4 (*Guo et al., 2024*), impairs development, suggesting functional redundancy. While recent studies shed light on ZGA initiation, the mechanisms responsible for extinguishing the 2c transcriptional program remain poorly understood.

Endogenous retroviruses (ERVs) are key regulators of early development. Murine endogenous retrovirus-L (MERVL) elements are transcriptionally active in 2c embryos (*Svoboda et al., 2004*) and drive chimeric transcripts that define the totipotent state (*Macfarlan et al., 2012*; *Peaston et al., 2004*). MERVL activation is required for blastocyst formation (*Sakashita et al., 2023*) and is mediated by DUX/OBOX and SRF binding at MT2 long terminal repeats (LTRs) (*Hermant et al., 2025*; *Yang et al., 2024*). However, sustained MERVL expression—such as that induced by forced DUX expression—disrupts totipotency exit and embryo development (*Guo et al., 2019*; *Percharde et al., 2018*). Although timely repression of MERVLs and the 2c program requires DUXBL (*Vega-Sendino et al., 2024*), the mechanisms by which DUXBL silences totipotency-related genes remain unknown.

Because genome-wide DNA demethylation occurs after fertilization, preimplantation embryos rely heavily on histone modifications to regulate transcription. During cleavage-stage development, H3K9me3 is deposited at retrotransposons; MERVL elements begin to acquire this repressive mark by the 2c stage, as DNA methylation levels decline (*Wang et al., 2018*). SETDB1, a key H3K9 methyltransferase, is thus a strong candidate for regulating MERVL silencing and early embryonic transcriptional control. In ESCs, SETDB1 represses transposable elements (TEs) via TRIM28 (KAP1)-dependent H3K9me3 deposition and by maintaining DNA methylation at LTRs and imprinted DMRs (*Karimi et al., 2011*; *Leung et al., 2014*; *Matsui et al., 2010*). However, its role at MERVLs in ESCs is debated: some studies report minimal repression (*Maksakova et al., 2013*), while others show that SETDB1 silences MERVLs to prevent dedifferentiation into 2CLCs (*Wu et al., 2020*). In oocytes, *Setdb1* loss leads to derepression of ERVK and ERVL-MaLR retrotransposons (*Eymery et al., 2016*; *Kim et al., 2016*). Whether SETDB1 also regulates MERVLs and totipotency exit in cleavage-stage embryos remains unresolved.

To address this question and understand why *Setdb1*[mat-/+] embryos fail to exit the cleavage stages, we employed a conditional knockout strategy. We previously showed that maternal SETDB1 protects the maternal pronucleus from TET3-mediated DNA demethylation in zygotes (*Zeng et al., 2019*). Here, we investigate its role in preimplantation development by analyzing the transcriptomes of 2c and rare 8c *Setdb1*[mat-/+] embryos. We find that maternally deposited SETDB1 is required to silence MERVL expression and 2c-specific totipotency transcripts, enabling successful progression beyond the cleavage stages.

## Results

### *Setdb1* maternal knockout embryos fail preimplantation development

A complete understanding of early embryonic development—and its intrinsic directionality and tempo—requires in vivo approaches. To address the in vivo role of SETDB1 in controlling the earliest stages of embryo development, we applied a genetic approach. We crossed *Setdb1*^f/f; *Zp3*-cre females with wild-type JF1/Ms males to obtain maternal knockout (*Setdb1*^mat-/+) (KO) embryos. Control *Setdb1*^f/f (WT) embryos were derived from *Setdb1*^f/f females lacking the *Zp3*-cre transgene. We previously reported that H3K9me2 and H3K9me3 histone marks are globally reduced in *Setdb1*^mat-/+ zygotes following *Zp3*-cre-driven excision of the SET domain from growing oocytes (*Zeng et al., 2019*).

The maternal *Setdb1* transcript is present in oocytes and persists throughout preimplantation development, while zygotic expression from the paternal allele begins only at the blastocyst stage (*Dodge et al., 2004*). Consequently, *Setdb1*^mat-/+ embryos lack SETDB1 protein function from the oocyte through the 8c stage.

At 1.5 dpc, over 70% of control eggs had reached the 2c stage, with a few progressing to the 4c stage (*Figure 1A*). In contrast, fewer than 30% of *Setdb1*^mat-/+ eggs reached the 2c stage, with many arrested at GV, MI, or MII stages. By 2.5 dpc, ~80% of control embryos reached the 8c stage, while only ~4% of mutant embryos did so. These findings agree with earlier studies (*Eymery et al., 2016*; *Kim et al., 2016*) and show that preimplantation development is severely impacted in *Setdb1*^mat-/+ embryos.

### Single-embryo total RNA sequencing of *Setdb1*^mat-/+ embryos

As no *Setdb1*^mat-/+ embryos develop into blastocysts, we performed single-embryo RNA sequencing at the 2c and 8c stages (1.5 and 2.5 dpc, respectively) to determine how maternal SETDB1 supports development beyond the cleavage stage. The KO embryos that did develop to the 2c or 8c stage were morphologically indistinguishable from WT under light microscopy (*Figure 1—figure supplement 1A*). To optimally assess both regular and chimeric transcription, we used total RNA sequencing. We profiled both KO and WT embryos with >6 biological replicates (*Figure 1*). Examples of normally expressing transcripts are displayed in *Figure 1—figure supplement 1B*.

Principal component analysis (PCA) revealed that samples clustered primarily by developmental stage (PC1), and secondarily by genotype (PC2) (*Figure 1B*), indicating that stage-specific transcriptional programs remain largely intact—more distinctly so in the rare 8c-stage embryos. We defined differentially expressed genes (DEGs) using thresholds of $|log_2FC|>1$ and adjusted $p<0.05$ across four pairwise comparisons (*Figure 1C*).

Volcano plots (*Figure 1D*) visualize the four key comparisons, and we provide the matching gene set enrichment analysis (GSEA) to each pairwise contrast (*Figure 1—figure supplement 2*): 'To be normal' at the 2c (2cWT vs. 2cKO) and 8c (8cWT vs. 8cKO) stages; and 'To develop' from 2c to 8c in WT (8cWT vs. 2cWT) and KO (8cKO vs. 2cKO) embryos. The volcano plots and an intersection analysis (*Figure 1E and F*) using a four-way Venn diagram approach revealed that most DEGs fell into the 'To develop' category, consistent with the PCA results.

### Transcriptional programs of development do not collapse in the absence of maternal SETDB1

We first examined DEGs associated with the 'To develop' category. We identified 1722 genes downregulated during development in both WT (8cWT vs. 2cWT) and KO (8cKO vs. 2cKO) embryos (*Figure 1E*), suggesting their normal downregulation is not dependent on SETDB1. This trend was confirmed by heatmap clustering (*Figure 1—figure supplement 3A*).

Gene ontology (GO) terms associated with these downregulated DEGs included DNA binding, transcriptional activation, ubiquitination, and phospholipid binding (*Figure 1—figure supplement 3B*). Similarly, 828 genes were consistently upregulated during development in both WT and KO embryos (*Figure 1F*) and were enriched in RNA production and translation (*Figure 1—figure supplement 3D*). Together, these results indicate that maternal SETDB1 is not required for the global execution of 2c-to-8c transcriptional transitions, and that the developmental arrest observed is not due to a complete transcriptional collapse.

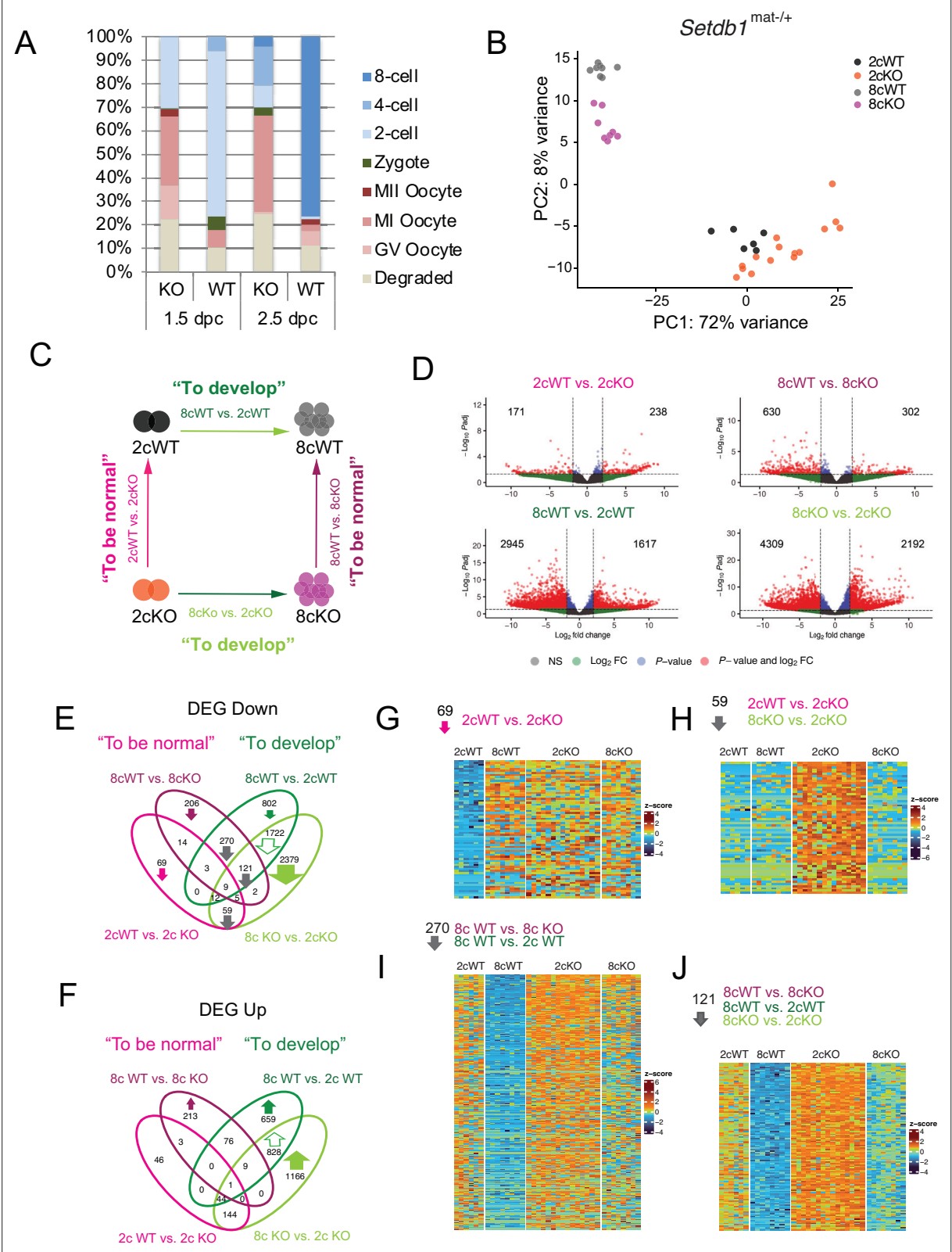

**Figure 1.** Maternal SETDB1 is essential for development beyond the eight-cell stage. (**A**) Quantification of *Setdb1*mat-/+ (KO) and *Setdb1*fl/+ (WT) embryo stages from the following number of total recovered embryos: KO (n=638), WT (n=484) at 1.5 dpc, and KO (n=310) and WT (n=80) at 2.5 dpc. (**B**) Principal component analysis of single-embryo total RNA-seq data from 2cWT (n=6), 2cKO (n=15), 8cWT (n=8), and 8cKO (n=8) embryos (*Figure 1—source data 1*). (**C**) Schematic of four pairwise comparisons defining requirements for normalcy and development. (**D**) Volcano plots highlighting

*Figure 1 continued on next page*

*Figure 1 continued*

differentially expressed genes (DEGs) using |log₂FC|>1 and adjusted p<0.05. (**E, F**) Four-way DEG comparisons (***Figure 1—source data 2***) visualized by Venn diagrams: (**E**) downregulated; (**F**) upregulated. (**G–J**) Heatmaps of DEGs from Venn compartments, showing stage- and genotype-specific patterns.

The online version of this article includes the following source data and figure supplement(s) for figure 1:

**Source data 1.** Sample information.

**Source data 2.** Differential gene expression (DGE) analysis.

**Figure supplement 1.** Normal features of *Setdb1* KO embryos.

**Figure supplement 2.** Gene set enrichment analysis (GSEA) of *Setdb1* KO differentially expressed genes (DEGs).

**Figure supplement 2—source data 1.** Gene set enrichment analysis (GSEA) on gene ontology (GO).

**Figure supplement 3.** Developmental transcriptional changes independent of maternal SETDB1.

**Figure supplement 3—source data 1.** Overrepresentation analysis (ORA) against gene ontology (GO) analysis by differentially expressed gene (DEG) Venn compartment.

**Figure supplement 4.** Developmental misregulation in *Setdb1* KO embryos.

**Figure supplement 5.** SETDB1 suppresses transposable elements at cleavage stages.

**Figure supplement 5—source data 1.** Differential TE expression (DTE): multimapping reads.

**Figure supplement 5—source data 2.** Differential TE expression (DTE): unimapping reads.

## The two-cell program is misregulated in the absence of maternal SETDB1

To understand which developmental change occurs specifically in *Setdb1*^mat-/+ embryos, we focused on genes differentially expressed only in KO embryos during the 2c to 8c transition. We identified 2379 downregulated and 1166 upregulated DEGs in the 8cKO vs. 2cKO contrast, but not in the WT developmental comparison (***Figure 1E and F***).

Heatmaps revealed that many genes exhibited aberrant expression in 2cKO embryos, with levels reverting to near-normal by the 8c stage (***Figure 1—figure supplement 4A and C***). Overrepresentation analysis (ORA) indicated that 2cKO embryos underexpressed genes related to ribosomal and mitochondrial function (***Figure 1—figure supplement 4B***), while overexpressing genes associated with later developmental processes such as pattern specification and organ morphogenesis (***Figure 1—figure supplement 4D***). These observations suggest that mis-timed activation or repression occurs in the absence of maternal SETDB1.

In a related heat map comparison (***Figure 1—figure supplement 4E***), we identified genes uniquely overexpressed in 2cKO embryos that are enriched in ribosomal and mitochondrial pathways (***Figure 1—figure supplement 4F***). Thus, maternal SETDB1 is required to properly regulate two-cell-stage gene expression dynamics.

## Maternal SETDB1 has a broad effect on transposable elements in 2c and 8c embryos

We next examined whether maternal SETDB1 regulates TE expression during preimplantation. PCA of TE expression in our RNA-seq data revealed clustering first by stage and then by genotype, with increased variability among 8cKO samples (***Figure 1—figure supplement 5A***). Multimapped (***Figure 1—figure supplement 5B***) and unimapped (***Figure 1—figure supplement 5—source data 2***) TE analysis showed that maternal SETDB1 is required to suppress TEs in WT vs. KO embryos and to control TEs during 2c to 8c development.

The tallies of 'To be normal' DE TEs by TE families show that several TE families—including ERVK, ERVL, ERVL-MaLR, ERV1, and L1 elements—require suppression by SETDB1 in WT embryos (***Figure 1—figure supplement 5C***). The ERVK, ERVL-MaLR, and ERV1 families were most affected in both 2cWT vs. 2cKO and 8cWT vs. 8cKO contrasts, while LINEs required SETDB1 for repression mainly at 2c. The tallies of 'To develop' suggest that SETDB also affects developmental TE changes during 2c-to-8c transitions. ERVK and LINE-1 elements appear to be developmentally activated in WT but less so in KO embryos (***Figure 1—figure supplement 5D***), which we investigate further below.

Heatmaps of DE TEs (***Figure 1—figure supplement 5E***) revealed strong derepression of nearly all repeat families in KO embryos at both the 2c and 8c stages. In WT embryos, L1 LINEs and many ERVKs

were upregulated between 2c and 8c, while MERVLs and ERVL-MaLRs were appropriately silenced (*Figure 1—figure supplement 5F*). This developmental regulation was disrupted in KO embryos: 8cKO vs. 2cKO showed partial MERVL/ERV1 repression and inappropriate activation of some ERVKs and LINEs—due to the highly derepressed state in 2cKO embryos.

These results collectively show that maternal SETDB1 is a major regulator of TE expression, with broad effects on multiple TE families including the 2c hallmark MERVL during early development.

## Maternal SETDB1 is required for timely suppression of cleavage-stage transcripts

To further explore SETDB1's role in establishing a normal 2c transcriptome, we looked at 2cWT vs. 2cKO DEGs and found 171 downregulated and 238 upregulated transcripts (*Figure 1E and F*). Among the 69 downregulated genes exclusive to 2cWT vs. 2cKO, many are prematurely activated in 2cKO embryos, including *Klf10*, *Lig1*, and *H1f3* (*Figure 1G*). Another 59 DEGs downregulated in 2cWT vs. 2cKO overlapped with 8cKO vs. 2cKO DEGs and were misexpressed only at the 2c stage (*Figure 1H*), suggesting these transcripts are normally repressed by SETDB1 at 2c but not at 8c. A notable example is *Zscan4c*, a key regulator of the 2c program in vitro (*Eckersley-Maslin et al., 2019*).

To understand SETDB1's role at the 8c stage, we inspected the 636 downregulated and 302 upregulated DEGs in 8cWT vs. 8cKO embryos (*Figure 1*). Of these, 270 overlapped with 8cWT vs. 2cWT DEGs (*Figure 1E*), indicating that maternal SETDB1 normally suppresses these transcripts during the 2c to 8c transition (*Figure 1I*). This group included *Usp17ld*, *Tdpoz1,* and *Tdpoz3*, transcripts expressed transiently at the 1c, 2c, or 4c stages (*Park et al., 2015*).

An additional 121 DEGs overlapped with both developmental and genotype comparisons, suggesting that their silencing occurs more slowly in *Setdb1*^mat-/+ embryos than in controls (*Figure 1J*). These included the 2c transient genes (*Park et al., 2015*) *Obox3* and *Fzd7*, further implicating maternal SETDB1 in regulating 2c-specific transcriptional shutdown. Taken together, this four-way comparison approach highlights maternal SETDB1 as a key factor for the timely suppression of specific transcripts at both the 2c and 8c stages.

## Maternal SETDB1 controls transient gene expression during ZGA

We next asked whether maternal SETDB1 regulates specific ZGA gene classes as defined in the 'Database of Transcriptome in Mouse Early Embryos' (DBTMEE) (*Park et al., 2015*). ORA (*Figure 2A*) using DBTMEE gene clusters from *Sakashita et al., 2023* revealed distinct patterns across four two-way DEG comparisons.

Maternal RNA and minor ZGA genes were appropriately downregulated in both WT and KO embryos from the 2c to 8c stage, as expected during normal development. However, 2c-specific transient transcripts were significantly downregulated in 8cWT vs. 2cWT embryos but not in 8cKO vs. 2cKO embryos. Moreover, 8cWT vs. 8cKO downregulated DEGs were enriched in this gene set, indicating a requirement for maternal SETDB1 to suppress 2c-transient transcripts by the 8c stage.

Similarly, 1c transient genes at the 2c stage and minor ZGA genes at the 8c stage were also insufficiently silenced in KO embryos. By contrast, mid-preimplantation gene activation (MGA) genes and their upstream regulators were upregulated normally in both genotypes. Notably, major ZGA and MGA transition-related genes were also underexpressed in 2cKO embryos, suggesting that SETDB1 may also support their proper induction.

Representative IGV browser tracks illustrate misregulation patterns in five biological replicates per condition (*Figure 2B*), including *Zscan4c* (minor ZGA), *Usp17ld* (1c transient) *Dbf4* (2c transient), and *Tdpoz3* (4c transient). Differential expression status is mapped across Venn diagram segments (*Figure 1E*), and selected transcripts are shown in boxplots (*Figure 2C*).

These results demonstrate that maternal SETDB1 is required to suppress 1c transient genes by the 2c stage, and minor ZGA and 2c-specific genes by the 8c stage, ensuring transcriptional fidelity across cleavage stages.

## Maternal SETDB1 suppresses two-cell-specific MERVL-chimeric transcripts

MERVL retroelements act as alternative promoters for chimeric transcripts during the 2c stage and serve as molecular hallmarks of totipotency (*Macfarlan et al., 2012*). Although SETDB1 represses

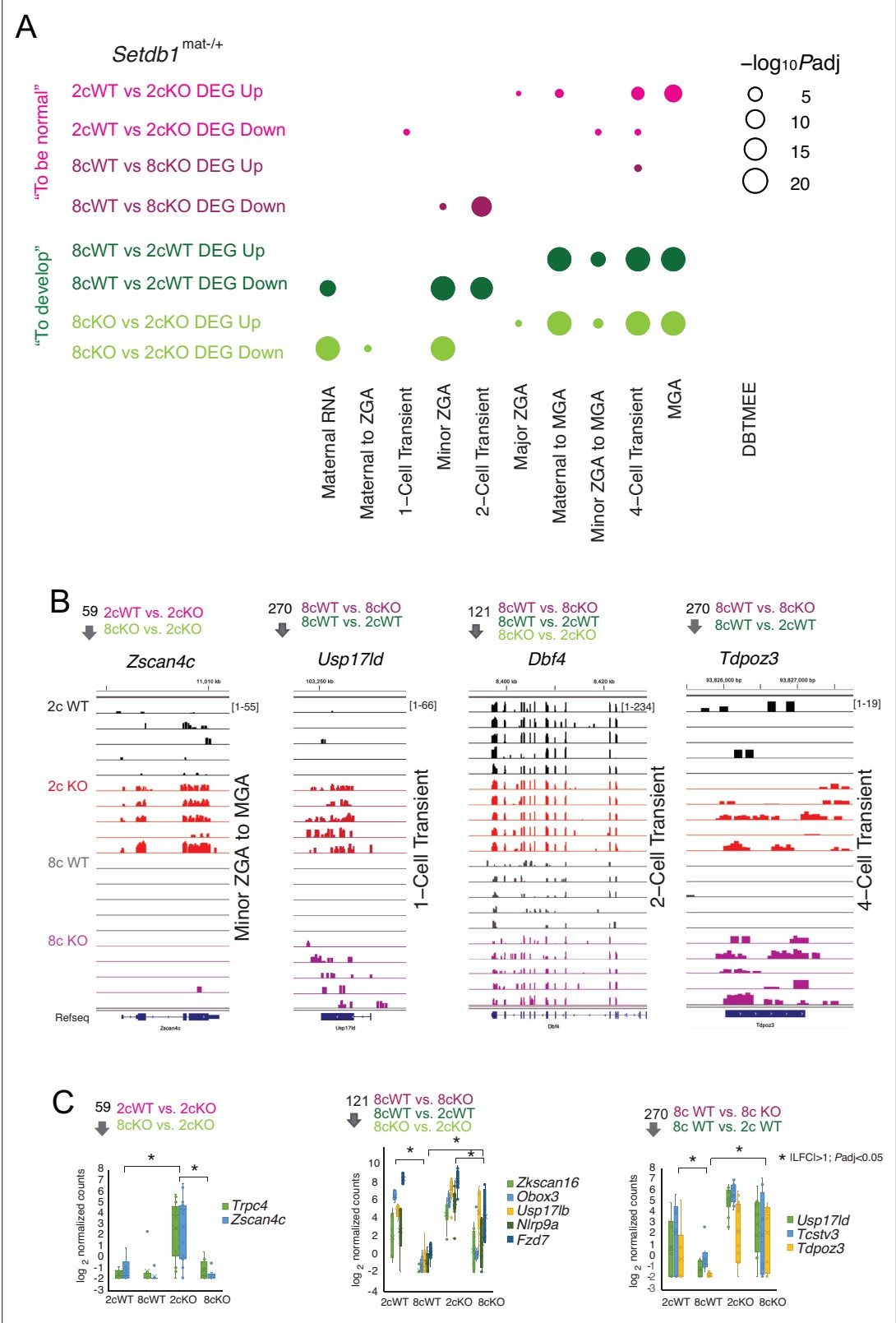

**Figure 2.** Maternal SETDB1 extinguishes two-cell transient gene expression. (**A**) Bubble plot showing overrepresentation analysis of Database of Transcriptome in Mouse Early Embryos (DBTMEE) (*Park et al., 2015*)-defined transcript sets among differentially expressed genes (DEGs) identified in the four pairwise comparisons (*Figure 2—source data 1*). (**B**) IGV browser snapshots of representative transcripts from the DBTMEE-defined minor ZGA to MGA, two-cell transient, and four-cell transient gene sets across five biological replicates. Venn diagram compartments are indicated above each

*Figure 2 continued on next page*

*Figure 2 continued*

track. Units in brackets represent normalized counts per million (CPM). Bigwig tracks are shown in the transcriptional direction matching the depicted gene. (**C**) Boxplots of selected DEGs *(|log₂FC|>1, adjusted p<0.05) from each Venn compartment, based on data from 2cWT (n=6), 2cKO (n=15), 8cWT (n=8), and 8cKO (n=8) embryos.

The online version of this article includes the following source data for figure 2:

**Source data 1.** Overrepresentation analysis (ORA) with Database of Transcriptome in Mouse Early Embryos (DBTMEE) categories.

ERVK and ERVL-MaLR retrotransposons in oocytes, it does not affect MERVLs at that stage (*Kim et al., 2016*).

We found significantly higher transcript levels of *MERVLint:ERVL:LTR* and *MT2_Mm:ERVL:LTR* in 2cKO than in 2cWT embryos (*Figure 3A*). These transcripts were efficiently extinguished in WT embryos by the 8c stage but remained partially expressed in KO embryos. The difference was even more pronounced at 8c: *MT2* expression was downregulated in 8cWT vs. 8cKO with LFC = −2.65 and p adj=1.31E–16, compared to LFC = −1.86 and p adj=1.41E–08 in 2cWT vs. 2cKO.

Similarly, many MERVL chimeric transcripts defined by *Macfarlan et al., 2012* were upregulated in both 2cKO and 8cKO embryos (*Figure 3B*), with most clustering on the left side of volcano plots (downregulated in WT vs. KO) (*Figure 3C and D*). Two shared DEGs reached significance at 2c, and nine at 8c (p adj <0.05, |log₂FC|>1). IGV views show misregulation of representative transcripts *Slc19a3* and *Trak2* (*Figure 3E*).

Using a combined list of previously identified (*Modzelewski et al., 2021*) and novel chimeric transcripts, we performed a four-way comparison to assess SETDB1 dependency (*Figure 3F*). SETDB1 had a strong suppressive effect on TE-driven, including MERVL-driven transcripts at both 2c and 8c stages, with no clear activating role. Thus, maternal SETDB1 is essential to silence 2c-specific MERVL chimeric transcripts during preimplantation development.

## Maternal SETDB1 silences MERVL MT2 LTR-activated transcripts

MT2s represent major MERVL subfamilies. Their LTRs can act as alternative promoters or distal enhancers for endogenous genes, depending on their genomic position and orientation. A recent study by *Yang et al., 2024* used a CRISPR-based epigenetic MT2 LTR inactivation system (MT2i) to identify MT2-controlled genes in mouse embryos. Using ChIP-seq data from *Wang et al., 2018* we found that H3K9me3 deposition at MT2-dependent transcripts is first reduced between E2c and L2c but later increases from 4c to the 8c stage (*Figure 4A*). MT2s that drive those transcripts accumulate H3K9me3 peaks progressively from the E2c to 8c stages (*Figure 4B*), and these peaks are stronger and more spatially defined. Peaks localized at the MT2 edges are consistent with low internal mappability in repeat elements. IGV browser views show that the expression of MT2-controlled DEGs such as *Arsk* and *Usp17ld* was derepressed in 2cKO and 8cKO embryos, while their flanking MERVLs and MT2s displayed strong H3K9me3 peaks in WT embryos (*Figure 4C–E*).

Heatmaps reveal that the expression of MT2-controlled DEGs in general was derepressed in 2cKO and 8cKO embryos (*Figure 4F*). These genes clustered on the left of volcano plots (*Figure 4G and H*), indicating strong SETDB1-mediated repression. *Yang et al., 2024* identified 341 genes requiring MT2 for activation by early 2c (E2c) and 1111 by late 2c (L2c). We observed that many of these genes were derepressed in SETDB1 KO embryos across time (*Figure 5A–E*). These results provide genetic evidence that maternal SETDB1 is essential for silencing MT2-activated genes.

## Maternal SETDB1 is required to extinguish DUXBL-responsive transcripts

DUXBL is a known repressor of ZGA and MERVL genes, acting in the footsteps of DUX during the exit from totipotency (*Vega-Sendino et al., 2024*). We tested whether maternal SETDB1 is required for DUXBL-mediated suppression. We compared our RNA-seq data to the top 50 DEGs from *Duxbl*⁻/⁻ embryos (*Vega-Sendino et al., 2024*) and found >2-fold derepression of 7 and 12 DUXBL-suppressed transcripts in 2cKO and 8cKO embryos, respectively (*Figure 6A*), suggesting that maternal SETDB1 acts specifically in their silencing. Volcano plots (*Figure 6B*) showed DUXBL-upregulated genes clustering on the left in both 2cWT vs. 2cKO and 8cWT vs. 8cKO comparisons, indicating again SETDB1-dependent suppression.

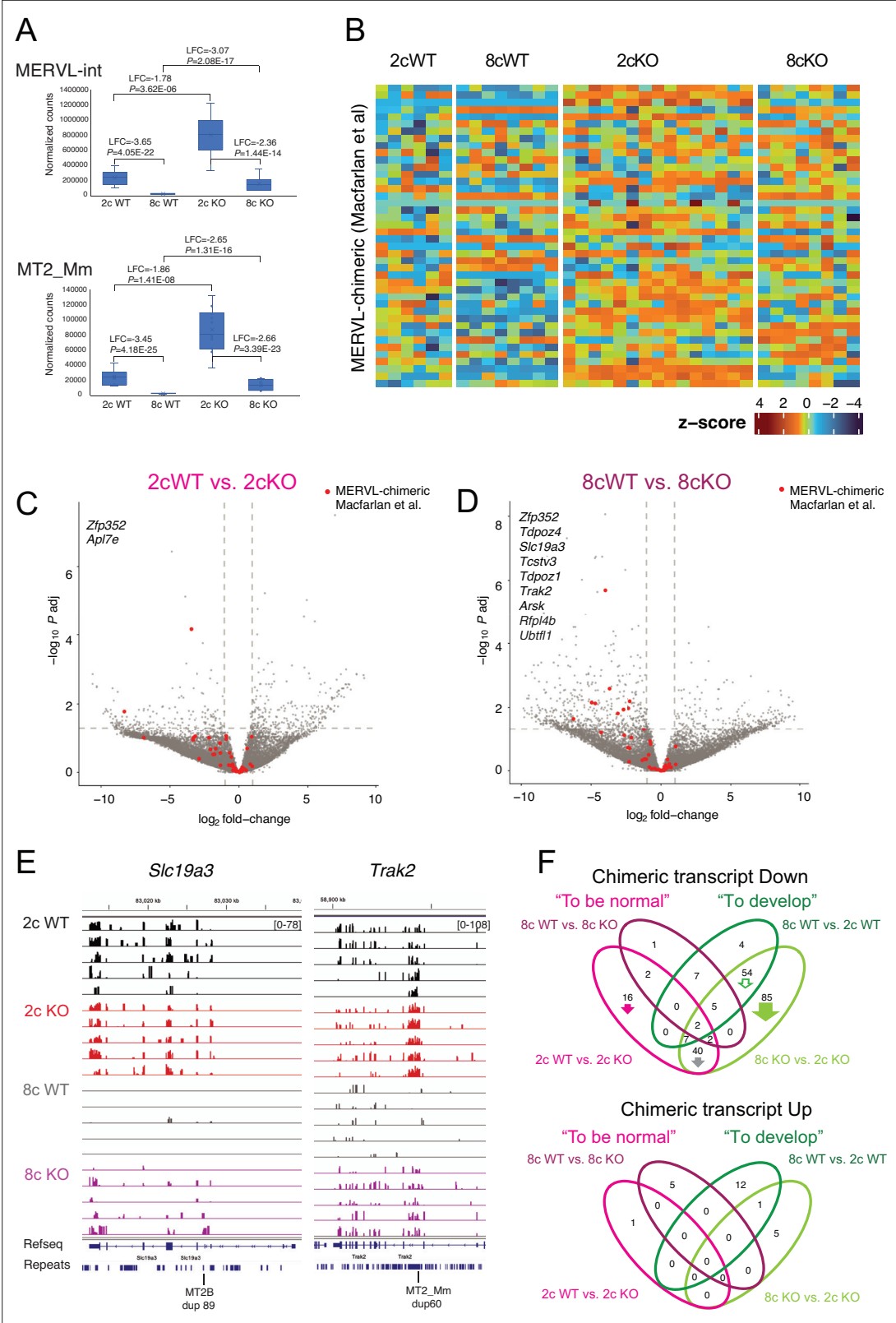

**Figure 3.** Maternal SETDB1 regulates MERVL-driven chimeric transcripts. (**A**) Boxplots showing normalized counts of multimapped MERVL-int and MT2_Mm elements. (**B**) Heatmap from *Setdb1* KO embryos at MERVL chimeric transcripts classified by ***Macfarlan et al., 2012***. (**C, D**) Volcano plots marking Macfarlan-defined chimeric transcripts in 2cWT vs. 2cKO and 8cWT vs. 8cKO pairwise comparisons. (**E**) IGV browser images of MT2B1 LTR-

*Figure 3 continued on next page*

*Figure 3 continued*

driven MERVL-chimeric transcripts. (**F**) Venn diagram showing differentially expressed (p<0.05) known and novel TE-driven chimeric transcripts (***Figure 3—source data 1***).

The online version of this article includes the following source data for figure 3:

**Source data 1.** Chimeric transcript analysis.

IGV browser views confirmed that *Dennd4c*, *Tmem132c* (***Figure 6D and E***), *Antxr1*, and *Aqr* (***Figure 6—figure supplement 1A and B***) were derepressed in KO embryos and associated with H3K9me3 peaks flanking MERVL elements in WT embryos. Although H3K9me3 signal within MERVLs is low due to mappability limits, adjacent flanks exhibited strong enrichment beginning at the 2c stage and increasing through 4c to morula (***Wang et al., 2018***). Two additional genes, *Nelfa* and *Zscan4d*, appeared derepressed in KO embryos (***Figure 6—figure supplement 1C and D***), despite not crossing significance thresholds. MERVLs serve as alternative promoters or distal enhancers for these transcripts, with H3K9me3 deposited near these elements in WT but not KO embryos.

Together, these data provide genetic evidence that DUXBL requires maternal SETDB1 to silence a subset of its targets via H3K9me3 deposition at MERVL LTRs.

## Overrepresentation analysis confirms shared targets with MT2 and DUXBL pathways

We performed ORA to test for overlap between SETDB1-sensitive genes and those misregulated by MT2i (***Yang et al., 2024***) or DUXBL KO (***Vega-Sendino et al., 2024***; ***Figure 7A***). We found highly significant overlap between proximal MT2-controlled DEGs and DEGs downregulated in 2cWT vs. 2cKO (p adj=3.19E–04); proximal MT2-controlled DEGs and DEGs in 8cWT vs. 8cKO (p adj=7.17E–04); E2c MT2i-downregulated genes and SETDB1-repressed DEGs at 2c (p adj=8.18E–05); and E2c MT2i-downregulated genes and SETDB1-repressed DEGs at 8c (p adj=5.77E–11). This suggests that genes activated by MT2s during ZGA also require maternal SETDB1 for their timely silencing later.

We also observed strong enrichment between DUXBL-suppressed genes and SETDB1-repressed DEGs in 8cWT vs. 8cKO embryos (p adj = 0.0071), providing genetic evidence for a collaboration between DUXBL and SETDB1 in extinguishing the 2c program.

In summary, these findings establish SETDB1, a H3K9 methyltransferase, as a broad-acting chromatin regulator in the preimplantation-stage embryo required for: silencing MERVL retroviral elements; extinguishing minor ZGA and transient totipotency-associated genes; and terminating transcription of MT2-activated and DUXBL-regulated transcripts at key regulatory elements in cleavage-stage embryos.

## Discussion

SETDB1 is essential for mammalian development due to its role in establishing transcriptionally repressive H3K9me3 marks. Our findings indicate that the maternal-effect lethality observed in *Setdb1*[mat-/+] preimplantation embryos is not due to a global transcriptional collapse, but rather the failure to silence specific transcriptional programs that must be extinguished for normal progression through, and beyond, the cleavage stages. Exit from the totipotency state is essential for embryonic development (***Guo et al., 2019***; ***Percharde et al., 2018***). While SETDB1 affects a broad range of genes and retroelements during preimplantation development, its specific role in silencing the totipotency program alone underscores its critical role as a chromatin regulator in early embryogenesis.

Maternal SETDB1 functions in a highly specific and dynamic manner. At the 2c stage, it suppresses 1c transient and minor ZGA transcripts; by the 8c stage, it silences 2c transient and MERVL-associated chimeric transcripts. In normal development, these silencing events coincide with dynamic H3K9me3 deposition at MERVLs and MT2 LTRs (***Figure 4***). In ESCs, SETDB1 prevents dedifferentiation into 2c-like cells by depositing H3K9me3 at or near totipotency genes (***Wu et al., 2020***), suggesting that a similar mechanism may operate during preimplantation embryo development in vivo. However, the rarity of *Setdb1*[mat-/+] 8c embryos limits our ability to directly confirm H3K9me3 depletion at these loci—a limitation of this study.

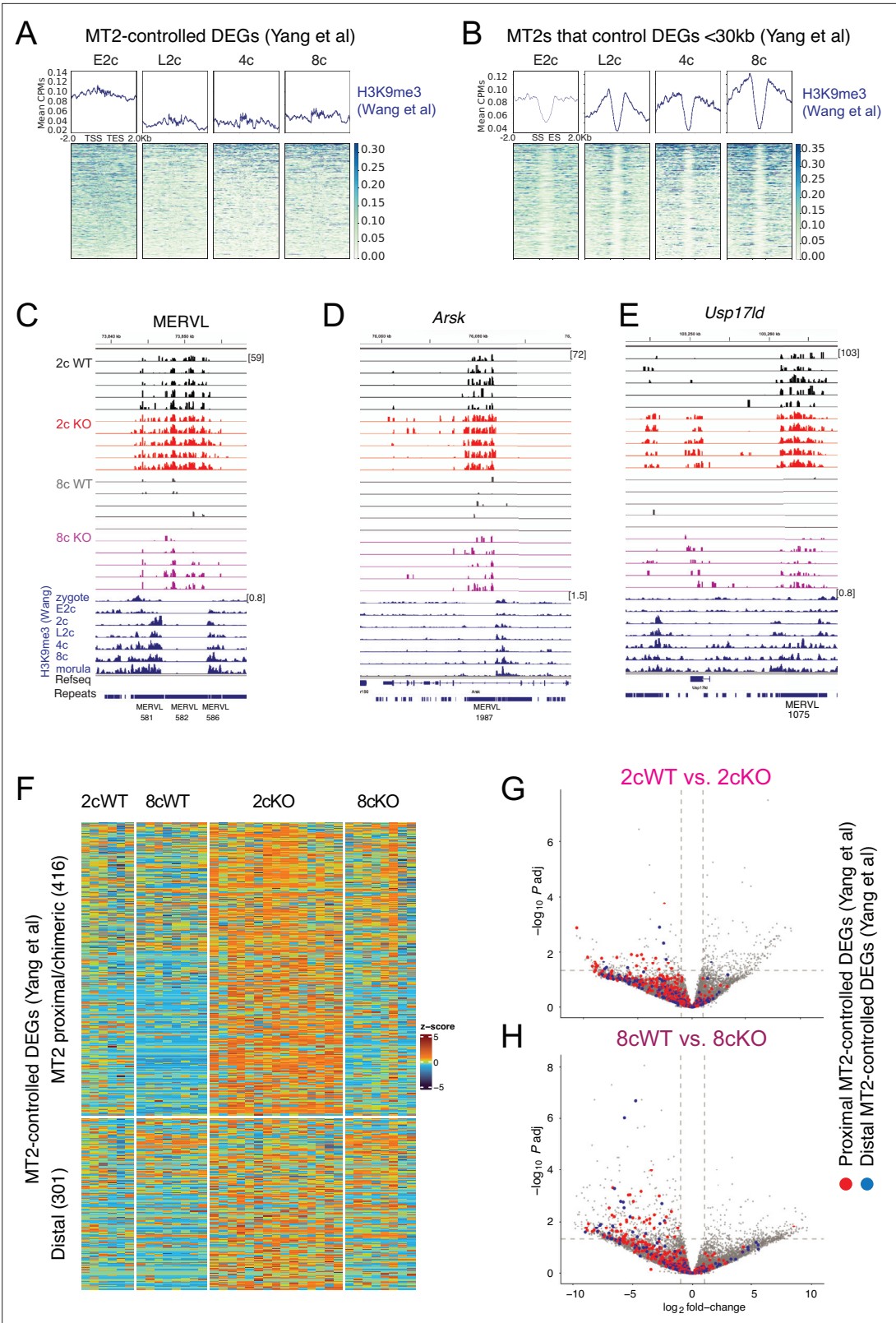

**Figure 4.** Maternal SETDB1 suppresses MT2 LTR-regulated genes. (**A, B**) Heatmaps of H3K9me3 deposition (*Wang et al., 2018*) at MT2-controlled (*Yang et al., 2024*) differentially expressed genes (DEGs) (**A**) and their MT2 elements (**B**) across early embryonic stages. (**C–E**) IGV browser views of a representative MERVL and MT2-regulated loci. (**F**) Heatmap of maternal *Setdb1* KO embryos at DEGs classified by MT2i data. (**G, H**) Volcano plots highlighting MT2-regulated genes in 2cWT vs. 2cKO and 8cWT vs. 8cKO pairwise comparisons.

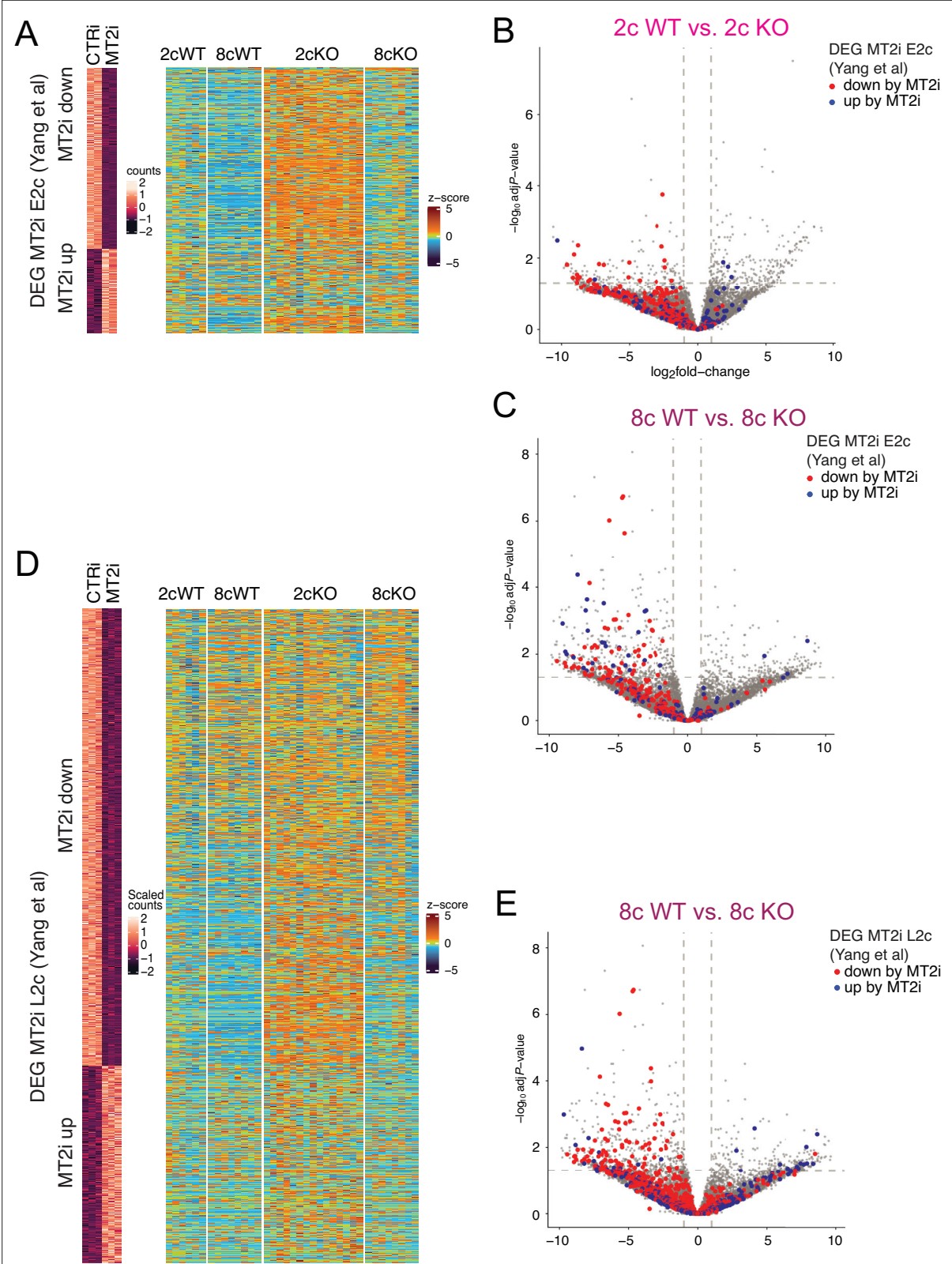

**Figure 5.** SETDB1 regulates MT2-activated genes across time. (**A**) Heatmaps comparing MT2i-responsive (*Yang et al., 2024*) early two-cell (E2c) differentially expressed genes (DEGs) with the *Setdb1* KO transcriptomes. (**B, C**) Volcano plots highlighting MT2-regulated E2c DEGs in 2cWT vs. 2cKO and 8cWT vs. 8cKO pairwise comparisons. (**D**) Heatmaps of MT2i late two-cell (L2c) DEGs. (**E**) Volcano plot highlighting MT2-regulated L2c DEGs.

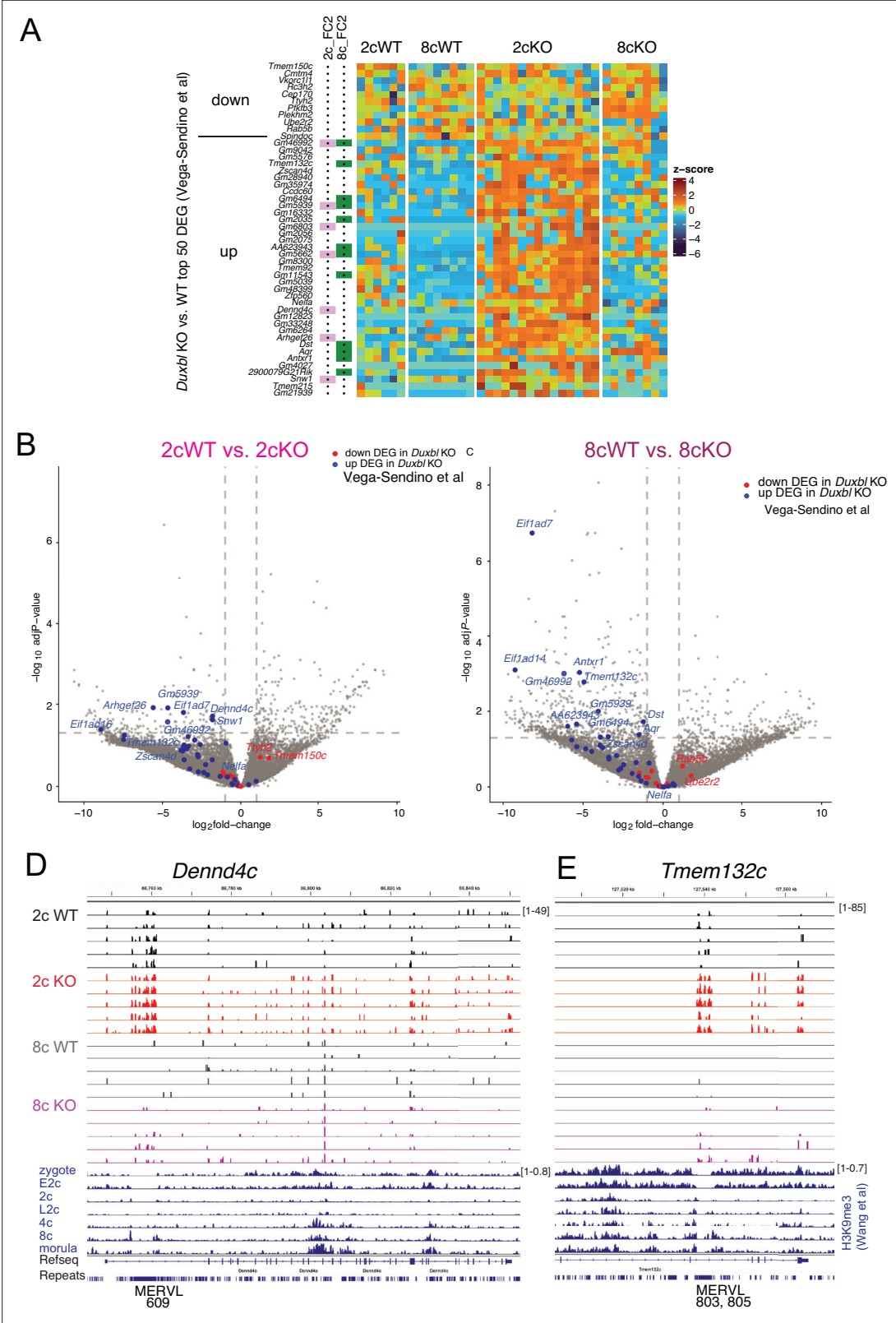

**Figure 6.** SETDB1 represses DUXBL-responsive transcripts. (**A**) Heatmap of top 50 differentially expressed genes (DEGs) identified in *Duxbl* KO embryos (***Vega-Sendino et al., 2024***) analyzed in the *Setdb1* KO RNA-seq dataset. (**B**) Volcano plots marking upregulated/downregulated DUXBL targets in 2cWT vs. 2cKO and 8cWT vs. 8cKO pairwise comparisons. (**D, E**) IGV browser examples of DUXBL-responsive DEGs with aligned H3K9me3 ChIP-seq data.

*Figure 6 continued on next page*

*Figure 6 continued*

The online version of this article includes the following figure supplement(s) for figure 6:

**Figure supplement 1.** Examples of DUXBL targets derepressed in *Setdb1* KO embryos.

SETDB1's mechanism likely involves recruitment to specific retroelement loci, such as MT2 LTRs by transcription factors, such as DUXBL. Our genetic data show that SETDB1 is required for silencing both MT2 LTRs and DUXBL-responsive transcripts. Detecting overlap of SETDB1-regulated genes with those affected by MT2 inactivation (*Yang et al., 2024*) and DUXBL loss (*Vega-Sendino et al., 2024*), along with the accumulation of H3K9me3 at these loci during normal preimplantation development, suggests a coordinated silencing mechanism in the embryo. Phenotypic parallels between *Setdb1*mat-/+ and *Duxbl*-/- embryos also support this model. While *Duxbl*-/- embryos arrest at the 4c stage (*Vega-Sendino et al., 2024*), a few *Setdb1*mat-/+ embryos progress to the 8c stage, suggesting that SETDB1 may act redundantly with, or downstream of, DUXBL.

In conclusion, we identify maternal SETDB1 as a critical in vivo epigenetic regulator that extinguishes totipotency-associated transcriptional programs, particularly those driven by retroelements and responsive to DUXBL. Its role at MT2 elements supports a model in which SETDB1 provides an epigenetic memory, silencing retroelements shortly after their activation by DUX and OBOX. This mechanism rapidly restricts the 2c transcriptional program, enabling the transition to pluripotency. By repressing MT2 LTRs, SETDB1 ensures successful progression beyond the cleavage stages and supports the earliest cell fate transitions in the mammalian embryo (*Figure 7B*).

# Materials and methods

## Key resources table

| Reagent type (species) or resource | Designation | Source or reference | Identifiers | Additional information |
|---|---|---|---|---|
| Gene (*Mus musculus*) | *Setdb1* | NCBI | Gene: 84505 | |
| Genetic reagent (*M. musculus*) | *Setdb1*tm1a(EUCOMM)Wtsi | European Mouse Mutant Archive | EMMA ID EM:04052 | *Skarnes et al., 2011* |
| Genetic reagent (*M. musculus*) | B6.Cg-Tg(Pgk1-flpo)10Sykr/J | The Jackson Laboratory | RRID:IMSR_JAX:011065 | *Wu et al., 2009* |
| Genetic reagent (*M. musculus*) | C57BL/6-Tg(Zp3-cre)93Knw/J | The Jackson Laboratory | RRID:IMSR_JAX:003651 | *de Vries et al., 2000* |
| Software, algorithm | StringTie2 | https://github.com/skovaka/stringtie2 | RRID:SCR_016338 | *Kovaka et al., 2019* |
| Software, algorithm | Trim Galore (v0.60) | https://github.com/FelixKrueger/TrimGalore | RRID:SCR_011847 | *Krueger, 2025 Martin, 2011* |
| Software, algorithm | edgeR (v4.4.1) | https://bioconductor.org/packages/edgeR/ | RRID:SCR_012802 | *Chen et al., 2016*; *McCarthy et al., 2012*; *Robinson et al., 2010* |
| Software, algorithm | ClusterProfiler (v4.14.4) | https://bioconductor.org/packages/clusterProfiler/ | RRID:SCR_016884 | *Wu et al., 2021* |
| Software, algorithm | STAR (v2.7.8) | https://github.com/alexdobin/STAR | RRID:SCR_004463 | *Dobin et al., 2013* |
| Software, algorithm | Samtools (v1.17) | https://github.com/samtools/samtools/ | RRID:SCR_002105 | *Li et al., 2009* |
| Software, algorithm | FeatureCounts (Subread v2.0.0) | https://subread.sourceforge.net/featureCounts.html | RRID:SCR_012919 | *Liao et al., 2014* |
| Software, algorithm | DESeq2 (v1.46.0) | https://github.com/thelovelab/DESeq2 | RRID:SCR_000154 | *Love et al., 2014* |
| Software, algorithm | BWA (v0.7.1) | https://github.com/lh3/bwa | RRID:SCR_010910 | *Li and Durbin, 2009* |
| Software, algorithm | deepTools2 v3.5.2 | https://github.com/deeptools/deepTools | RRID:SCR_016366 | *Ramírez et al., 2016* |
| Software, algorithm | ggplot2 | https://ggplot2.tidyverse.org | RRID:SCR_014601 | *Wickham, 2016* |

*Continued on next page*

*Continued*

| Reagent type (species) or resource | Designation | Source or reference | Identifiers | Additional information |
|---|---|---|---|---|
| Software, algorithm | ComplexHeatmap (R) | https://jokergoo.github.io/ComplexHeatmap-reference/book/ | RRID:SCR_017270 | *Gu et al., 2016* |
| Software, algorithm | ggvenn (v0.1.10) | https://github.com/yanlinlin82/ggvenn | RRID:SCR_025300 | *Yan, 2023* |
| Software, algorithm | TEtranscripts (v2.2.3) | https://github.com/mhammell-laboratory/TEtranscripts | RRID:SCR_015687 | *Jin et al., 2015* |
| Chemical compound, drug | CARD HyperOva | Cosmo Bio | Cat. No. KYD-010-EX | |
| Commercial assay or kit | SMART-Seq Stranded Kit | Takara Biosciences USA, Mountain View CA | Cat. No. 634444 | |
| Commercial assay or kit | RNA-Bee | Amsbio | Cat. No. CS-105B | |
| Peptide, recombinant protein | RNasin ribonuclease inhibitor | Promega | Cat. No. N2515 | |
| Commercial assay or kit | DNA-free Kit | Ambion | Cat. No. AM1906 | |
| Commercial assay or kit | Agilent High Sensitivity DNA Kit | Agilent Technologies, Inc. | Part Number:5067-4626 | |
| Commercial assay or kit | QuantiFluor dsDNA System | Promega Corp., Madison, WI, USA | Cat. No. E2671 | |
| Commercial assay or kit | Kapa Illumina Library Quantification qPCR assays | Kapa Biosystems | Cat. No. KR0405 | |

## Mice

All animal experiments were conducted in accordance with the National Research Council's Guide for the Care and Use of Laboratory Animals, under Institutional Animal Care and Use Committee-approved protocol (AUP ref.no. 23-12-032) at the Van Andel Institute (VAI). The *Setdb1* conditional knockout mouse line (*Setdb1*$^{f/f}$) was derived from *Setdb1*$^{tm1a(EUCOMM)Wtsi}$ (*Skarnes et al., 2011*) (European Mouse Mutant Archive) as previously described (*Zeng et al., 2019*). The *Pgk*-neo cassette was removed via FLPE-mediated recombination using B6.Cg-Tg$^{(Pgk1-flpo)10Sykr/J}$ (*Wu et al., 2009*). *Zp3*-cre transgenic mice C57BL/6-Tg$^{(Zp3-cre)93Knw/J}$ (*de Vries et al., 2000*) were used to excise SET-domain–encoding exons in growing oocytes. *Setdb1*$^{f/f}$; *Zp3*-cre females were experimental animals; *Setdb1*$^{f/f}$ females without the transgene served as controls.

## Collection of preimplantation mouse embryos

Embryos were collected at late 2c (L2c) and 8c stages. For 2c embryos, 6–8-week-old females were superovulated with 5 IU PMSG followed by 5 IU hCG after 46–48 h, then mated with WT males. For 8c embryos, CARD HyperOva was used in 26–30-day-old females to increase efficiency. Embryos were harvested in M2 medium at 44 h (1.5 dpc, 2c) and 68 h (2.5 dpc, 8c) post-hCG. Embryos were washed in PBS, placed into 1.5 mL tubes, lysed in RNA-Bee, flash frozen, and stored at –80°C.

## RNA isolation

RNA was isolated from single embryos using RNA-Bee. Linear polyacrylamide (LPA) was used for carrier-mediated isopropanol precipitation. RNasin (Promega) was added during resuspension. DNA contamination was removed using rDNase I (Ambion). RNA was recovered after DNase inactivation and ethanol precipitation and eluted in RNasin-containing Tris buffer.

## Library preparation and sequencing

Libraries were prepared with the SMART-Seq Stranded Kit (Takara). Samples were randomized and processed with fragmentation to 250 bp (85°C, 10 min embryo RNA or 6 min control RNA). Unique dual indexes were added during PCR1 (10 cycles), and eight samples were pooled for PCR2 (12 cycles). Libraries were QC'd using Agilent chips, QuantiFluor, and KAPA qPCR, and sequenced on a NovaSeq S2 (paired-end 50 bp, ~25M reads/library).

## RNA-seq analysis

Reads were trimmed using Trim Galore (v0.60) (*Krueger, 2025*; *Martin, 2011*) and aligned to mm10 with STAR (v2.7.8) (*Dobin et al., 2013*) using `--quantMode GeneCounts`. DGE analysis was

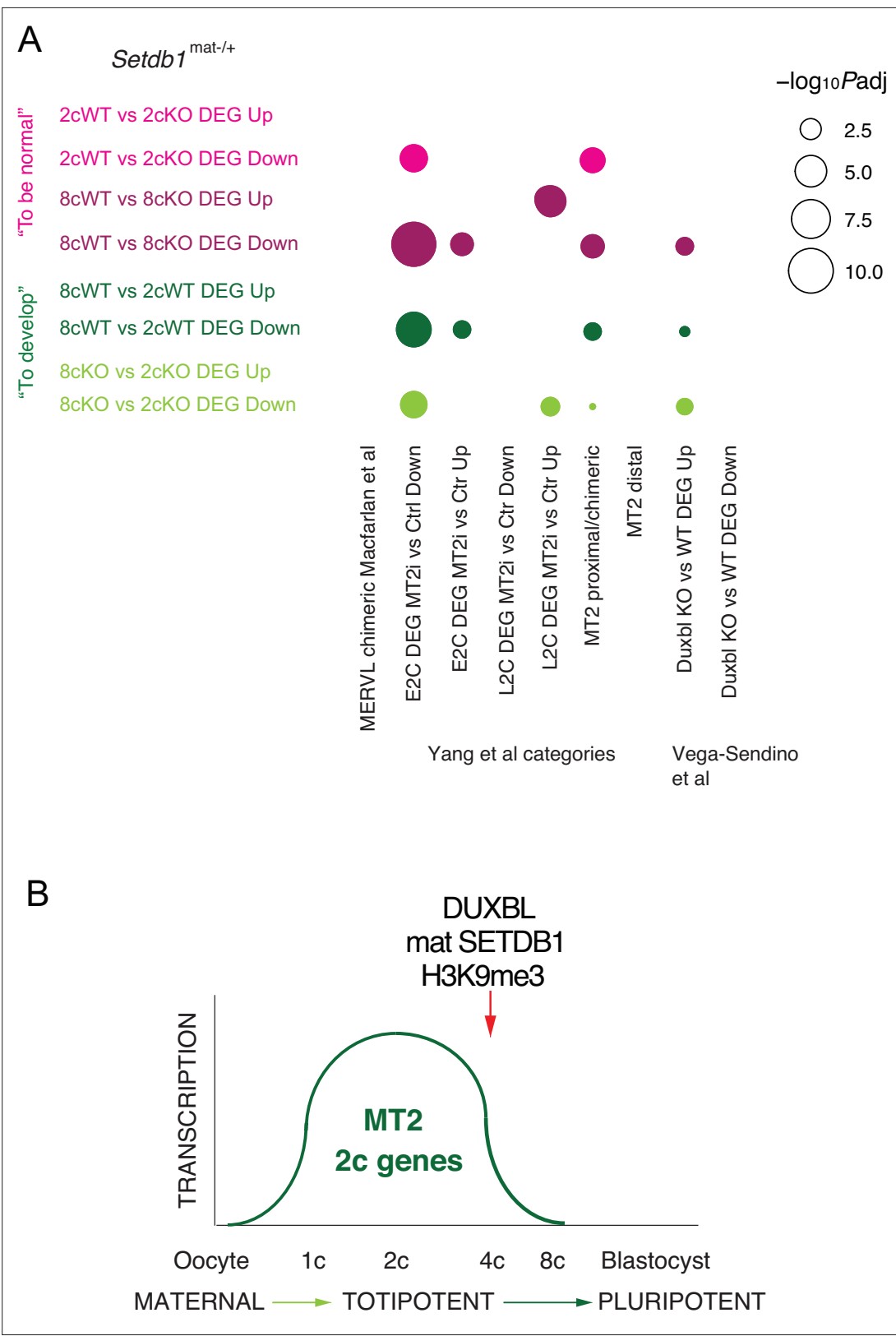

**Figure 7.** SETDB1 collaborates with DUXBL to repress totipotency programs. (**A**) Bubble plot of overrepresentation analysis showing enrichment of previously identified MERVL-chimeric, MT2i-responsive, and *Duxbl* KO-responsive gene sets (***Macfarlan et al., 2012***; ***Vega-Sendino et al., 2024***; ***Yang et al., 2024***) in *Setdb1* KO pairwise differentially expressed genes (DEGs) (***Figure 7—source data 1***). (**B**) Summary model: Maternal SETDB1 deposits H3K9me3 at MT2 elements to silence MERVL-driven two-cell transcripts in coordination with DUXBL, enabling exit from totipotency.

*Figure 7 continued on next page*

*Figure 7 continued*

The online version of this article includes the following source data for figure 7:

**Source data 1.** ORA with MERVL-chimeric, MT2i, and *Duxbl* KO gene sets.

performed with edgeR (v4.4.1) (*Chen et al., 2016*; *McCarthy et al., 2012*; *Robinson et al., 2010*). Briefly, gene counts were filtered to remove lowly expressed genes and normalized using the trimmed mean of M-values (TMM) method. Gene-wise dispersion was calculated and differential expression was assessed using a negative binomial generalized linear model in edgeR. P-values were adjusted for multiple testing using Benjamini–Hochberg adjustment (adjusted p<0.05). Normalized $\log_2$ counts were used in figures. Bigwig tracks were created using bamCoverage (deepTools v3.5.2, CPM normalization) (*Ramírez et al., 2016*). GSEA and ORA against GO and custom gene sets were performed with ClusterProfiler (v4.14.4) (*Wu et al., 2021*).

### Transposable element (TE) analysis

Two sets of data were analyzed for TE: multimapped reads and uni-mapped reads. For multimapped reads, STAR (v2.7.8) (*Dobin et al., 2013*) was run with `--winAnchorMultimapNmax 200`, `--outFilterMultimapNmax 100`, and `--outSAMmultNmax 1`. TEtranscripts (v2.2.3) (*Jin et al., 2015*) was used for multi-mapped TE counts. Uni-mapped reads were filtered with -q 255 using Samtools (v1.17) (*Li et al., 2009*). FeatureCounts (Subread v2.0.0) (*Liao et al., 2014*) was used with mm10_rmsk_TE.gtf to generate count matrices. Differential expression was assessed using DESeq2 (v1.46.0) (*Love et al., 2014*).

### Chimeric transcript analysis

Chimeric transcripts were quantified with Retrotransposon (v37), which uses STAR (*Liao et al., 2014*) and StringTie2 (*Kovaka et al., 2019*). Count tables were processed using DESeq2 (*Love et al., 2014*) for DGE (adjusted p<0.05).

### Public datasets

ChIP-seq fastq files (*Wang et al., 2018*) (GSE98149) were trimmed using Trim Galore (v0.60) (*Krueger, 2025*; *Martin, 2011*), aligned to mm10 with BWA (v0.7.1) (*Li and Durbin, 2009*), and filtered with Samtools (v1.17) (*Li et al., 2009*). BAMs were converted to BigWig and visualized with deepTools (*Ramírez et al., 2016*). Supplementary data was used from other studies (*Eymery et al., 2016*; *Macfarlan et al., 2012*; *Park et al., 2015*; *Vega-Sendino et al., 2024*; *Yang et al., 2024*), for comparative analyses.

### Data visualization

Heatmaps were generated with ComplexHeatmap (R) (*Gu et al., 2016*), bubble plots were generated using ggplot2 (*Wickham, 2016*), and Venn diagrams using ggvenn (v0.1.10) (*Yan, 2023*).

## Acknowledgements

We gratefully acknowledge the support from the Van Andel Institute (VAI) Genomics Core (RRID:SCR_022913), the VAI Bioinformatics and Biostatistics Core (RRID:SCR_024762), and the VAI Vivarium (RRID:SCR_023211). This work was supported by the Van Andel Institute.

## Additional information

### Funding

| Funder | Grant reference number | Author |
| --- | --- | --- |
| Van Andel Institute | | Piroska E Szabó |

The funders had no role in study design, data collection and interpretation, or the decision to submit the work for publication.

## Author contributions
Tie-Bo Zeng, Ji Liao, Investigation, Writing – review and editing; Zhen Fu, Formal analysis, Visualization, Writing – original draft, Writing – review and editing; Mary F Majewski, Marie Adams, Investigation, Methodology, Writing – review and editing; Piroska E Szabó, Conceptualization, Formal analysis, Supervision, Funding acquisition, Investigation, Methodology, Writing – original draft, Project administration, Writing – review and editing

## Author ORCIDs
Tie-Bo Zeng ⓘ https://orcid.org/0000-0002-8999-2952
Zhen Fu ⓘ https://orcid.org/0000-0002-8120-590X
Marie Adams ⓘ https://orcid.org/0000-0001-7909-2339
Piroska E Szabó ⓘ https://orcid.org/0000-0001-9314-7009

## Ethics
All animal experiments were conducted in accordance with the National Research Council's Guide for the Care and Use of Laboratory Animals, under Institutional Animal Care and Use Committee-approved protocol (AUP ref.no. 23-12-032) at the Van Andel Institute (VAI).

Reviewer #1 (Public review): https://doi.org/10.7554/eLife.109248.2.sa1
Reviewer #2 (Public review): https://doi.org/10.7554/eLife.109248.2.sa2
Author response https://doi.org/10.7554/eLife.109248.2.sa3

# Additional files

## Supplementary files
MDAR checklist

## Data availability
RNA sequence data were deposited in GEO (accession number GSE269417). All data generated or analyzed during this study are included in the manuscript and supporting files; source data files have been provided for all figures.

The following previously published datasets were used:

| Author(s) | Year | Dataset title | Dataset URL | Database and Identifier |
|---|---|---|---|---|
| Wang C, Liu X, Gao Y, Yang L, Li C, Liu W, Chen C, Kou X, Zhao Y, Wang H, Zhang Y, Gao S | 2018 | Reprogramming of H3K9me3-dependent heterochromatin during mammalian early embryo development [ChIP-seq] | https://www.ncbi.nlm.nih.gov/geo/query/acc.cgi?acc=GSE98149 | NCBI Gene Expression Omnibus, GSE98149 |
| Szabó PE, Zeng T, Fu Z | 2025 | Single embryo total RNA sequencing to map the effect of maternal SETDB1 on the 2-celll and 8-cell transcriptome | https://www.ncbi.nlm.nih.gov/geo/query/acc.cgi?acc=GSE269417 | NCBI Gene Expression Omnibus, GSE269417 |

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
