## [Editor Report · eLife Assessment]

This study presents a **valuable** finding on maternal SETDB1 as a key chromatin repressor that shuts down the 2C gene program and enables normal mouse embryonic development. The evidence supporting the claims of the authors is **solid**, although the inclusion of a causality test, a mechanistic understanding of SETDB1 targeting, and phenotypic quantification would have greatly strengthened the study. The work will be of broad interest to biologists working on embryonic development, stem cells and gene regulation.

---

## [Referee Report · Reviewer #1 (Public review)]

Summary:

During the earliest stages of mouse development, the zygote and 2-cell (2C) embryo are totipotent, capable of generating all embryonic and extra-embryonic lineages, and they transiently express a distinctive set of "2C-stage" genes, many driven by MERVL long terminal repeat (LTR) promoters. Although activation of these transcripts is a normal feature of totipotency, they must be rapidly silenced as development proceeds to the 4-cell and 8-cell stages; failure to shut down the 2C program results in developmental arrest. This study examines the role of maternal SETDB1, a histone H3K9 methyltransferase, in suppressing the 2C transcriptional network. Using an oocyte-specific conditional knockout that removes maternal Setdb1 while leaving the paternal allele intact, the authors demonstrate that embryos lacking maternal SETDB1 arrest during cleavage, with very few progressing beyond the 8-cell stage and no morphologically normal blastocysts forming. Transcriptomic analyses reveal persistent expression of MERVL-LTR-driven transcripts and other totipotency markers, indicating a failure to terminate the totipotent state. Together, the data demonstrate that maternally deposited SETDB1 is required to silence the MERVL-driven 2C program and enable the transition from totipotency to pluripotency. More broadly, the work identifies maternal SETDB1 as a key chromatin repressor that deposits repressive H3K9 methylation to shut down the transient 2C gene network and to permit normal preimplantation development.

Strengths:

(1) Closes a key knowledge gap.

The study tackles a central open question - how embryos exit the totipotent 2-cell (2C) state - and provides direct in vivo evidence that epigenetic repression is required to terminate the 2C program for development to proceed. By identifying maternal SETDB1 as the responsible factor, the work substantially advances our understanding of the maternal-to-zygotic transition and early lineage specification.

(2) Clean genetics paired with rigorous genomics.

An oocyte-specific Setdb1 knockout cleanly isolates a maternal-effect requirement, ensuring that early phenotypes arise from loss of maternal protein. The resulting cleavage-stage arrest is unambiguous (most embryos stall before or around the 8-cell stage). State-of-the-art single-embryo RNA-seq across stages - well-matched to low-cell-number constraints - captures genome-wide mis-expression, including persistent 2C transcripts in mutants, strongly supporting the conclusions.

(3) Compelling molecular linkage to phenotype.

Transcriptome data show that without maternal SETDB1, embryos fail to repress a suite of 1-cell/2C-specific genes by the 8-cell stage. The tight correlation between continued activation of the MERVL-driven totipotency network and developmental arrest provides a specific molecular explanation for the observed failure to progress.

(4) Mechanistic insight grounded in chromatin biology.

SETDB1, a H3K9 methyltransferase classically linked to heterochromatin and transposon repression, targets MERVL LTRs and MERVL-driven chimeric transcripts in early embryos. Bioinformatic evidence indicates that these loci normally acquire H3K9me3 during the 2C→4C transition. The data articulate a coherent mechanism: maternal SETDB1 deposits repressive H3K9me3 at 2C gene loci to shut down the totipotency network, extending observations from ESC systems to bona fide embryos.

(5) Broad implications for development and stem-cell biology.

By pinpointing a maternal gatekeeper of the totipotent-to-pluripotent transition, the work suggests that some cases of cleavage-stage arrest (e.g., in IVF) may reflect faulty epigenetic silencing of transposon-driven genes. It also informs stem-cell efforts to control totipotent-like states in vitro (e.g., 2C-like cells), linking epigenetic reprogramming, transposable-element regulation, and developmental potency.

Weaknesses:

(1) Causality not directly demonstrated.

The link among loss of SETDB1, persistence of 2C transcripts, and developmental arrest is compelling but remains correlative. No rescue experiments test whether dampening the 2C/MERVL program restores development. Targeted interventions-e.g., knocking down key 2C drivers (such as Dux) or pharmacologically curbing MERVL-linked transcription in maternal Setdb1 mutants-would strengthen the claim that unchecked 2C activity is causal rather than a by-product of other SETDB1 functions.

(2) Limited mechanistic resolution of SETDB1 targeting.

The study establishes a requirement for maternal SETDB1 but does not define how it is recruited to MERVL loci. Given SETDB1's canonical cooperation with TRIM28/KAP1 and KRAB-ZNFs, upstream sequence-specific factors and/or pre-existing chromatin features likely guide targeting. Direct occupancy and mark-placement evidence (e.g., SETDB1/TRIM28 CUT&RUN or ChIP, and H3K9me3 profiling at MERVL LTRs during the 2C→4C window) would convert inferred mechanisms into demonstrated ones.

(3) Narrow scope on MERVL; broader epigenomic consequences underexplored.

Maternal SETDB1 may restrain additional repeat classes or genes beyond the 2C network. A systematic repeatome analysis (LINEs/SINEs/ERV subfamilies) would clarify specificity versus a general loss of heterochromatin control. Moreover, potential effects on imprinting or DNA methylation balance are not examined; perturbations there could also contribute to arrest. Bisulfite-based DNA methylation maps at imprinted loci and allele-specific expression analyses would help rule in/out these mechanisms.

(4) Phenotype quantitation and transcriptomic breadth could be clearer.

The developmental phenotype is described qualitatively ("very few beyond 8-cell") without precise stage-wise arrest rates or representative morphology. Tabulated counts (2C/4C/8C/blastocyst), images, and statistics would increase clarity. On the RNA-seq side, the narrative emphasizes known 2C markers; reporting novel/unannotated misregulated transcripts, as well as downregulated pathways (e.g., failure to activate normal 8-cell programs, metabolism, or early lineage markers), would present a fuller portrait of the mutant state.

---

## [Referee Report · Reviewer #2 (Public review)]

Zeng et al. report that Setdb1-/- embryos fail to extinguish the 1- and 2-cell embryo transcriptional program and have permanent expression of MERVL transposable elements. The manuscript is technically sound and well performed, but, in my opinion, the results lack conceptual novelty.

(1) The manuscript builds on previous observations that: 1, Setbd1 is necessary for early mouse development, with knockout embryos rarely reaching the 8-cell stage; 2, SETB1 mediates H3K9me3 deposition at transposable elements in mouse ESCs; 3, SETB1silences MERVLs to prevent 2CLC-state acquisition in mouse ESCs. The strength of the current work is the demonstration that this is not due to a general transcriptional collapse; but otherwise, the findings are not surprising. The well-known (several Nature papers of years ago) crosstalk between m6A RNA modification and H3K9me3 in preventing 2CLC generation also partly compromises the novelty of this work.

(2) The conclusions regarding H3K9me3 deposition are inferred based on previously reported datasets, but there is no direct demonstration.

(3) The detection of chimeric transcripts is somewhat unreliable using short-read sequencing.

---

## [Author Response]

**eLife Assessment**
This study presents a valuable finding on maternal SETDB1 as a key chromatin repressor that shuts down the 2C gene program and enables normal mouse embryonic development. The evidence supporting the claims of the authors is solid, although the inclusion of a causality test, a mechanistic understanding of SETDB1 targeting, and phenotypic quantification would have greatly strengthened the study. The work will be of broad interest to biologists working on embryonic development, stem cells and gene regulation.

Thank you for this positive evaluation of our work. Please find the point-by point responses to the Reviewer’s comments below.

**Public Reviews:**

**Reviewer #1 (Public review):**
Summary:During the earliest stages of mouse development, the zygote and 2-cell (2C) embryo are totipotent, capable of generating all embryonic and extra-embryonic lineages, and they transiently express a distinctive set of "2C-stage" genes, many driven by MERVL long terminal repeat (LTR) promoters. Although activation of these transcripts is a normal feature of totipotency, they must be rapidly silenced as development proceeds to the 4-cell and 8-cell stages; failure to shut down the 2C program results in developmental arrest. This study examines the role of maternal SETDB1, a histone H3K9 methyltransferase, in suppressing the 2C transcriptional network. Using an oocyte-specific conditional knockout that removes maternal Setdb1 while leaving the paternal allele intact, the authors demonstrate that embryos lacking maternal SETDB1 arrest during cleavage, with very few progressing beyond the 8-cell stage and no morphologically normal blastocysts forming. Transcriptomic analyses reveal persistent expression of MERVL-LTR-driven transcripts and other totipotency markers, indicating a failure to terminate the totipotent state. Together, the data demonstrate that maternally deposited SETDB1 is required to silence the MERVL-driven 2C program and enable the transition from totipotency to pluripotency. More broadly, the work identifies maternal SETDB1 as a key chromatin repressor that deposits repressive H3K9 methylation to shut down the transient 2C gene network and to permit normal preimplantation development.Strengths:(1) Closes a key knowledge gap.The study tackles a central open question - how embryos exit the totipotent 2-cell (2C) state - and provides direct in vivo evidence that epigenetic repression is required to terminate the 2C program for development to proceed. By identifying maternal SETDB1 as the responsible factor, the work substantially advances our understanding of the maternal-to-zygotic transition and early lineage specification.(2) Clean genetics paired with rigorous genomics.An oocyte-specific Setdb1 knockout cleanly isolates a maternal-effect requirement, ensuring that early phenotypes arise from loss of maternal protein. The resulting cleavage-stage arrest is unambiguous (most embryos stall before or around the 8-cell stage). State-of-the-art single-embryo RNA-seq across stages - well-matched to low-cell-number constraints - captures genome-wide mis-expression, including persistent 2C transcripts in mutants, strongly supporting the conclusions.(3) Compelling molecular linkage to phenotype.Transcriptome data show that without maternal SETDB1, embryos fail to repress a suite of 1-cell/2C-specific genes by the 8-cell stage. The tight correlation between continued activation of the MERVL-driven totipotency network and developmental arrest provides a specific molecular explanation for the observed failure to progress.(4) Mechanistic insight grounded in chromatin biology.SETDB1, a H3K9 methyltransferase classically linked to heterochromatin and transposon repression, targets MERVL LTRs and MERVL-driven chimeric transcripts in early embryos. Bioinformatic evidence indicates that these loci normally acquire H3K9me3 during the 2C→4C transition. The data articulate a coherent mechanism: maternal SETDB1 deposits repressive H3K9me3 at 2C gene loci to shut down the totipotency network, extending observations from ESC systems to bona fide embryos.(5) Broad implications for development and stem-cell biology.By pinpointing a maternal gatekeeper of the totipotent-to-pluripotent transition, the work suggests that some cases of cleavage-stage arrest (e.g., in IVF) may reflect faulty epigenetic silencing of transposon-driven genes. It also informs stem-cell efforts to control totipotent-like states in vitro (e.g., 2C-like cells), linking epigenetic reprogramming, transposable-element regulation, and developmental potency.

We thank Reviewer 1 for recognizing the strengths in our work and for the suggestions below.

Weaknesses:(1) Causality not directly demonstrated.The link among loss of SETDB1, persistence of 2C transcripts, and developmental arrest is compelling but remains correlative. No rescue experiments test whether dampening the 2C/MERVL program restores development. Targeted interventions-e.g., knocking down key 2C drivers (such as Dux) or pharmacologically curbing MERVL-linked transcription in maternal Setdb1 mutants-would strengthen the claim that unchecked 2C activity is causal rather than a by-product of other SETDB1 functions.

We agree that rescue experiments might strengthen causality. Those experiments, however, would be extremely challenging technically because the knockdowns would need to be precisely timed to follow (and not prevent) the wave of 2c-specific activation. Knocking down 2c drivers in the zygote, for example, may prevent switching on the totipotency program. In addition, while sustained MERVL expression—such as that induced by forced DUX expression—disrupts totipotency exit and embryo development (1, 2), derepression of transcription is very broad in Setdb1^mat-/+^ embryos and knocking down individual 2C drivers may not be sufficient to rescue development or restore the exit from totipotency.

(2) Limited mechanistic resolution of SETDB1 targeting.The study establishes a requirement for maternal SETDB1 but does not define how it is recruited to MERVL loci. Given SETDB1's canonical cooperation with TRIM28/KAP1 and KRAB-ZNFs, upstream sequence-specific factors and/or pre-existing chromatin features likely guide targeting. Direct occupancy and mark-placement evidence (e.g., SETDB1/TRIM28 CUT&RUN or ChIP, and H3K9me3 profiling at MERVL LTRs during the 2C→4C window) would convert inferred mechanisms into demonstrated ones.

We do show H3K9me3 patterns at MERVL LTRs during the early2c-late2c-2c-4c-8c-morula window from a published dataset. Please see the genome browser images in Figures 4C, 4D, 4E, 6D, 6E and Figure S6. We agree that mapping of SETDB1/TRIM28 to those locations would strengthen the mechanistic insight. However, ChIPseq or CUT&RUN of those proteins in preimplantation embryos are not technically feasible. We do provide genetic evidence for the collaboration between SETDB1 and DUXBL, a DNA-binding factor, by showing that DUXBL cannot switch off its top targets without SETDB1 (Figure 6). Future studies will characterize the molecular mechanisms underlying this (likely indirect) collaboration. We do not think that DUXBL and SETDB1 directly interact, because such interaction was not detected by DUXBL IP-MS (3).

(3) Narrow scope on MERVL; broader epigenomic consequences underexplored.Maternal SETDB1 may restrain additional repeat classes or genes beyond the 2C network. A systematic repeatome analysis (LINEs/SINEs/ERV subfamilies) would clarify specificity versus a general loss of heterochromatin control. Moreover, potential effects on imprinting or DNA methylation balance are not examined; perturbations there could also contribute to arrest. Bisulfite-based DNA methylation maps at imprinted loci and allele-specific expression analyses would help rule in/out these mechanisms.

We did examine genes and repeat elements beyond the 2c network. We evaluated gene and TE expression changes using four-way comparisons. Please find the results regarding gene expression in Figure 1C-J, Figure S2, Figure S3, Figure S4., Table S2, Table S3, and Table S4. Please find results on TE expression in Figure S5. Table S6, Table S7, and Table S8 and in the text. We agree that DNA methylation may be altered in Setdb1^mat-/+^ embryos. In our hands, evaluating this possibility using bisulfite sequencing requires a larger number of embryos than what we can feasibly obtain (the number of obtained mutant embryos is very small). Regarding imprinted gene expression, one cannot fully assess and interpret imprinted gene expression in preimplantation stage embryos before the maternally deposited transcripts are gone. We reported earlier that clear somatic parental-specific patterns of imprinted gene expression may only start later in development, around 8.5 dpc (4).

(4) Phenotype quantitation and transcriptomic breadth could be clearer.The developmental phenotype is described qualitatively ("very few beyond 8-cell") without precise stage-wise arrest rates or representative morphology. Tabulated counts (2C/4C/8C/blastocyst), images, and statistics would increase clarity. On the RNA-seq side, the narrative emphasizes known 2C markers; reporting novel/unannotated misregulated transcripts, as well as downregulated pathways (e.g., failure to activate normal 8-cell programs, metabolism, or early lineage markers), would present a fuller portrait of the mutant state.

Tabulated counts are displayed in Figure 1A, and morphology is shown in Figure S1A. We do say that 4% Setdb1^mat-/+^ embryos reached the 8-cel stage by 2.5 dpc. We recovered zero Setdb1^mat-/+^ blastocysts at 4.5 dpc (not shown). On the RNA-seq side we do report a more global assessment of transcription of genes and TEs (please see above at point 3), including novel chimeric transcripts (Table S6). Developmental pathways are shown in Figure S3 and Figure S4. Metabolic pathways are displayed in Figure S2.

**Reviewer #2 (Public review):**
Zeng et al. report that Setdb1-/- embryos fail to extinguish the 1- and 2-cell embryo transcriptional program and have permanent expression of MERVL transposable elements. The manuscript is technically sound and well performed, but, in my opinion, the results lack conceptual novelty.(1) The manuscript builds on previous observations that: 1, Setbd1 is necessary for early mouse development, with knockout embryos rarely reaching the 8-cell stage; 2, SETB1 mediates H3K9me3 deposition at transposable elements in mouse ESCs; 3, SETB1silences MERVLs to prevent 2CLC-state acquisition in mouse ESCs. The strength of the current work is the demonstration that this is not due to a general transcriptional collapse; but otherwise, the findings are not surprising. The well-known (several Nature papers of years ago) crosstalk between m6A RNA modification and H3K9me3 in preventing 2CLC generation also partly compromises the novelty of this work.

We thank the Reviewer for appreciating the technical quality of our work. Regarding novelty, please consider that prior work in ES cells included contradictory findings (please see our Introduction). Prior embryology work (please see our Introduction) did not explain the preimplantation-stage phenotype. We highly appreciate those earlier works. Our work here answers the expectations drawn from prior studies and unequivocally shows that SETDB1 carries out the developmentally essential function of suppressing MERVLs and the 2-cell program in the mouse embryo.

(2) The conclusions regarding H3K9me3 deposition are inferred based on previously reported datasets, but there is no direct demonstration.

Dynamic H3K9me3 deposition is displayed at MERVL LTRs during the early2c-late2c-2c-4c-8c-morula window (Figures 4C, 4D, 4E, 6D, 6E and Figure S6) from a published work that has very high-quality data. We agree that demonstrating loss off H3K9me3 in Setdb1^mat-/+^ embryos would confirm that the H3K9me3 histone methyltransferase function of SETDB1 (as opposed to any, yet unidentified, non-HMT specific activity of SETDB1) is responsible for shutting down MERVL LTRs. However, ChIP-seq, CUT&RUN, or similar assays are not feasible due to the rarity of Setdb1^mat-/+^ embryos.

(3) The detection of chimeric transcripts is somewhat unreliable using short-read sequencing.

We used single embryo total RNA-seq and we report detecting chimeric transcripts (Table S6), which is considered more reliable than mRNA-seq for detecting chimeric transcripts, because many are not polyadenylated. We acknowledge, however, that long-read sequencing, which recently is becoming available, but which is still very expensive, is currently the most powerful method for detecting chimeric transcripts. This, however, does not affect the major conclusions or the significance of our work.